# Isolated Gastric Metastases of Pancreatic Ductal Adenocarcinoma following Radical Resection—Impact of Endosonography-Guided Fine Needle Aspiration Tract Seeding

**DOI:** 10.3390/biomedicines10061392

**Published:** 2022-06-12

**Authors:** Martin Loveček, Pavel Skalický, Ondřej Urban, Jana Tesaříková, Martin Kliment, Róbert Psár, Hana Švébišová, Kateřina Urban, Beatrice Mohelníková-Duchoňová, Dušan Klos, Martin Stašek

**Affiliations:** 1Department of Surgery I, University Hospital Olomouc, I.P. Pavlova 185/6, 77900 Olomouc, Czech Republic; martin.lovecek@fnol.cz (M.L.); pavel.skalicky@fnol.cz (P.S.); jana.tesarikova@fnol.cz (J.T.); dusan.klos@fnol.cz (D.K.); 2Department of Internal Medicine II—Gastroentrology and Geriatrics, University Hospital Olomouc, I.P. Pavlova 185/6, 77900 Olomouc, Czech Republic; ondrej.urban@fnol.cz; 3Helios Kliniken Schwerin, Klinik für Gastroenterologie und Infektiologie, Wismarsche Strasse 393-397, 19055 Schwerin, Germany; martin_kliment@hotmail.com; 4Department of Radiology, Faculty of Medicine and Dentistry, Palacky University Olomouc, Hněvotínská 976/3, 77900 Olomouc, Czech Republic; psar.robert@gmail.com; 5Department of Oncology, University Hospital Olomouc, I.P. Pavlova 185/6, 77900 Olomouc, Czech Republic; hana.svebisova@fnol.cz; 6Institute of Molecular and Translational Medicine, Faculty of Medicine and Dentistry, Palacký University, Olomouc, Hněvotínská 1333/5, 77900 Olomouc, Czech Republic; katerina.kolarova@upol.cz (K.U.); beatricemohelnikova.duchonova@fnol.cz (B.M.-D.); 7Department of Oncology, Faculty of Medicine and Dentistry, Palacký University, Hněvotínská 976/3, 77900 Olomouc, Czech Republic

**Keywords:** pancreatic adenocarcinoma, endoscopic ultrasound, fine-needle aspiration biopsy, needle tract seeding, gastric metastasis

## Abstract

Background: Endosonography-guided fine needle aspiration biopsy (EUS-FNA)-associated metachronous gastric seeding metastases (GSM) of pancreatic ductal adenocarcinoma (PDAC) represent a serious condition with insufficient evidence. Methods: Retrospective analysis of PDAC resections with a curative-intent, proven pathological diagnosis of PDAC, preoperative EUS-FNA and post-resection follow-up of at least 60 months. The systematic literature search of published data was used for the GSM growth evaluation using Pearson correlation and the linear regression analyses. Results: The inclusion criteria met 59/134 cases, 16 (27%) had retained needle tract (15 following distal pancreatectomy, 1 following pylorus-sparing head resection). In total, 3/16 cases (19%) developed identical solitary GSM (10–26th month following primary surgery) and were radically resected. A total of 30 published cases of PDAC GSM following EUS-FNA were identified. Lesion was resected in 20 distal pancreatectomy cases with complete information in 14 cases. A correlation between the metastasis size and time (r = 0.612) was proven. The regression coefficient b = 0.72 expresses the growth of 0.72 mm per month. Conclusions: The GSM represent a preventable and curable condition. A remarkably high number of GSM following EUS-FNA was identified, leading to follow-up recommendation of EUS-FNA sampled patients. Multimodal management (gastric resection, adjuvant chemotherapy) may prolong survival.

## 1. Introduction

Tissue confirmation of pancreatic ductal adenocarcinoma (PDAC) can be carried out using minimally invasive methods, such as percutaneous abdominal sampling (PAS) or endoscopically guided fine needle aspiration (EUS-FNA) [1]. EUS-FNA is considered accurate (sensitivity 85–92%; specificity 96–99%) and safe diagnostic method for verification of malignant cells in pancreatic solid lesions [1,2,3,4,5,6]. The reported complication rate of EUS-FNA is low (0.98–1.03%) [7,8]. The data on long-term complications including tumor seeding are rare and not congruent [1,9]. Regardless of the extremely rare occurrence of EUS-FNA-associated seeding metastases, they belong to late serious EUS-FNA-associated complications that may decrease the individual survival [7]. Thus far, only one multicentric Japanese study with six cases [10] and twenty-four case reports referring to needle tract seeding metastases of PDAC following EUS-guided sampling have been reported worldwide until 2022 [1,3,7,8,11,12,13,14,15,16,17,18,19,20,21,22,23,24,25,26,27,28,29,30]. Moreover, only 20 cases of those documented seeding PDAC metastases have been solved surgically with resection so far [3,7,8,10,13,19,21,22,23,24,25,26,27,28,29,30]. 

The aim was to analyze a cohort of PDAC patients who underwent curative-intent surgery with previous EUS-FNA verification and summarize all the data regarding EUS-FNA-associated seeding PDAC metastases. According to the guidelines (ESMO—European Society for Medical Oncology; S3—German guidelines), tissue confirmation is considered unnecessary for resectable PDAC. Clinical practice is still different and most resectable PDAC are still referred to surgery following EUS-FNA. 

## 2. Materials and Methods

Retrospective analysis of a prospectively maintained single-center database of PDAC patients operated on with curative intent (2010–2014), who underwent EUS-FNA before the surgery. All the data in the database had been collected prospectively, including tumor type, tumor location, stage, type of surgical procedure, oncological treatment, disease-free survival (DFS), recurrence location, additional treatment and overall survival (OS). The inclusion criteria of the study were as follows: (1) a curative-intent surgical treatment; (2) histopathological diagnosis of PDAC; (3) a preoperative EUS-FNA diagnostic procedure; and (4) post-resection follow-up comprising biochemical tumor markers monitoring (CA 19-9, CEA, CA 125) every 3 months, and imaging-computed tomography (CT) or positron emission tomography/computed tomography (PET/CT) scans performed every 6–12 months or in the case of CA 19-9 elevation. The exclusion criterion was extragastric recurrence (liver, peritoneum, lymph nodes, locoregional, lung, or multiple). All tissue samples (primary tumor, EUS-FNA samples and resected stomach wall with metastases) were verified by two independent pathologists. Seeding PDAC metastases (gastric wall recurrence) of pancreatic cancer were defined as histologically proven recurrent pancreatic cancer located in the area corresponding to the prior EUS-FNA channel, usually in a gastric wall. DFS was measured as the period between the date of surgery and the diagnosis of cancer recurrence. The OS was measured as the period between the date of surgery and the date of death.

Curative-intent surgery for body and tail tumor localization was distal pancreatectomy with splenectomy and standard lymphadenectomy according to International Study Group for Pancreatic Surgery (ISGPS) and the pylorus-preserving hemipancreatoduodenectomy (Traverso modification) with standard lymphadenectomy for periampullary or head localization. All EUS-FNA with 22-gauge needles were carried out by three experienced endosongraphists with 2–4 needle passes.

The Pearson correlation analysis and linear regression analysis of data from cases presented thus far and of our patients were used to evaluate the growth rate and time. The IBM SPSS Statistics version 22 (IBM, Armon, New York, USA) was used to analyze the data. The study has been approved by the Institutional Ethical Committee, corresponding ethical approval code 159/16. 

## 3. Results

The analysis identified 59 (44%) PDAC cases with preoperatively performed EUS-FNA from the database of 134 patients (Table 1). The group consisted of 44 following pancreas head resections—43 with a resected area of previous EUS-FNA during the curative-intent surgery and 15 patients following distal pancreatectomy (27%), in whom the retained needle tract area remained in the gastric wall (1 in a group of head location and 15 in a group of body/tail location). Fourteen of them (14/16) survived more than 1 year without radiologically proven recurrence. In three (19%) patients from this group (No = 16), an unusual location of metachronous oligometastases (gastric wall without serosal involvement) was found (Figure 1, Figure 2 and Figure 3). From the histopathological point of view, primary pancreatic tumor and secondary resected metastases in the stomach/pylorus have identical histopathological findings (Table 1).

The location of the primary tumor in the body/tail of the pancreas and primary procedure in two patients were distal pancreatectomies with splenectomy and lymphadenectomy. The third one was a unique case of recurrence following pylorus-preserving hemipancreatoduodenectomy, since no such case has been published previously. Radical resections of seeding metastases with uneventful recovery were completed in all three cases. The patients’ demographics, clinical characteristics and their treatment are summarized in Table 2. None of them had an early recurrence of the disease (during the first 6 months after the seeding metastasis resection).

Literary research and subsequent evaluation pointed out 30 published cases of PDAC GSM. Complete information for the progression analysis of seeded tumors was gained in 14 cases, as shown in Figure 4. All cases of seeding/needle tract metastases are illustrated in Table 3. There was a moderate positive correlation between size and time (r = 0.612). The regression coefficient b = 0.72 is significantly non-zero (*p* = 0.020) and expresses an increase by 0.72 mm in one month.

## 4. Discussion

Seeding following FNA is classified as a long-term and potentially relevant complication. In a retrospective study by Micames et al., peritoneal carcinomatosis in patients with pancreatic cancer is lower, when sampling is performed with the EUS-FNA (2.2%) vs. percutaneous FNA (16.3%) [31]. Needle tract metastases following EUS-FNA present only a very limited number of case reports. However, the fear of seeding is clearly illustrated in the clinical transplantation protocol of Mayo Clinic for the treatment of proximal cholangiocarcinoma [32]. A biopsy of the primary tumor excludes such patients from neoadjuvant therapy and liver transplantation due to a high rate of peritoneal metastasis [32,33]. With the increasing role of neoadjuvant therapy in the treatment of resectable and borderline resectable pancreatic carcinomas, EUS-FNA plays a crucial role in the diagnostic workup in these cases [34]. Despite EUS-FNA sensitivity and specificity reaching 85–89% and 96–99%, the actual guidelines for the management of primary radically operable pancreatic cancer (European Society of Gastrointestinal Endoscopy guidelines, German guidelines—S3) consider EUS-FNA as a non-mandatory method in the management of these cases [1,35,36]. Seeding is considered an overlooked and underestimated problem with clinical impact for the selected group of patients [37]. Current clinical practice is still not following recommendations and guidelines, and most resectable PDAC cases are referred to surgery following EUS-FNA. Kim et al. focused on peritoneal recurrence in a cohort of 411 cases. EUS-FNA was not associated with an increased rate of peritoneal recurrence, decrease in cancer-free survival or overall survival among PDAC patients [2]. However, seeding PDAC metastases (gastric recurrence) were probably missing. The PIPE study concluded that the EUS-FNA of IPMN was not associated with an increased frequency in peritoneal seeding in patients who underwent resection [38]. Despite that, international consensus guidelines (2012) do not recommend cyst fluid analysis and aspiration in mucinous-like pancreatic cystic lesions due to the real risk of peritoneal dissemination [39]. In his review, Minaga et al. (2017) present an increase in the number of case reports with the topic of gastric wall seeding metastases after the EUS-FNA among PDAC patients [15] and the number is still increasing [23,24,25,26,27,28,29,30]. Most of these reports come from Japan [3,8,11,12,13,14,15,19,21,22,23,24,25,26,27,28,30]. Only a number of the reported patients—23 among 30 cases of reported seeding metastases of radically resected PDAC—were subsequently resected with curative intent [3,8,10,13,17,21,22,23,24,25,26,27,28,29,30] (Table 3). In the presented group (radical surgery for PDAC 2011–2014 in our institution), the most frequent isolated PDAC metastases treated with curative intent surgery were just gastric metastases, followed by solitary pulmonary oligometastases [40]. According to El Hajj [41], it is very difficult to specify the real clinical risk of seeding of EUS-FNA among PDAC. According to a multicentric analysis of Yane, this clinical situation may exceed to 3.8% [10]. 

Since the PDAC is a highly lethal malignancy with a very low long-term survival rate, the real rate can only be considered among a specific subgroup of long-term survivors (only around 20% of all PDAC patients who underwent curative-intent surgery reach the 5-year survival) [41,42]. Systemic multiple recurrence (locoregional or/with liver, peritoneal or pulmonary) causes 30% lethality in the first and another 30% during the second postoperative year, respectively. Gastric needle tract metastases could be unidentified due to tumor biology. Based on the current documented cases, we have proposed a general characteristic and criteria of the group of patients, in whom true seeding metastases were evaluated.

### 4.1. Pathologically Confirmed Primary Diagnosis of the PDAC after Pancreatic Resection 

The aggressivity of PDAC is high even in the early stages. In our cohort, the cases were of stage I in one and stage II in two. The reported cases comprised stage I in 7/14 (50%) and stage II in 5/14 (35%) following TNM classification, 7th edition. No lymph node involvement except one and no distant metastases were detected (Table 3). The stage was not reported in six patients. The early stage is considered a favorable prognostic factor in the PDAC-resected patients in our cohort [40]. The first of our cases with N1 status underwent EUS-FNA 22 months prior to surgery. The final stage is supposed to be higher than the periprocedural one. The delay was caused by adverse post-EUS-FNA events and the initial refusal of surgery. 

### 4.2. Body/Tail Location of the Primary Tumor Is the Most Frequent

Resectable tumors are mostly located in the head of the pancreas. If the EUS-FNA is performed preoperatively, the needle channel is commonly in the duodenum and is usually removed during hemipancreatoduodenectomy. The needle channel is not only resected in cases of non-standard sampling using the needle tract through the gastric antrum or the pylorus. The pylorus-preserving procedure does not remove the needle tract with possible subsequent seeding, as first described in our study. The puncture tract is usually not resected in pancreatic neck and tail tumors, thus enabling the seeding recurrence [3,7,8,10,11,12,13,14,16,17,18,19,21,22,24,25,26,27,28,29,30]. The pyloric location of gastric metachronous seeding metastasis presented in this study is unique and first published as a result of EUS-FNA. The report of Yang et al. presents a gastric/pyloric metastasis of PDAC of probably hematogenous etiology. In this case, surgical therapy with curative intent was not provided [43].

### 4.3. Preoperative EUS-FNA

The EUS-FNA for PDAC is considered abundant in HR-CT proven resectability. In S3-leitlinien (German guidelines for resectable pancreatic exocrine tumors and European Society of Gastrointestinal Endoscopy guidelines), the mandatory diagnostic tools include abdominal ultrasound and EUS and the multi-detector high-resolution CT [1,9,35].

Current EUS-FNA indication reveals borderline resectable and locally advanced tumors, in which histopathological diagnosis is needed for the indication/initiation of neoadjuvant therapy. Despite this fact, almost half of resected patients in our study underwent EUS-FNA (N = 59/134).

### 4.4. 1-Year Survival without Other Recurrences

The evolution of pancreatic cancer and progression lasts over 10 years [44,45]. The datation of potential seeding and growth progression of malignant cells requires EUS-FNA, CT or PET/CT scans and the resected specimen size.

The metastases in our cohort were diagnosed in the second postoperative year, reaching the size of 18–33 mm. The studies focused on EUS-FNA long-term complications, covering only 3 months after the procedure [5]. Seeding tumor progression takes approximately 20 months to grow to a 2 cm tumor (median DFS 22.5 ± 10.6 months; Table 2). There is a good chance to diagnose seeding metastases in the curable stage with longer follow-up and a 3-month interval, as shown in Figure 4 and Table 3. There was a moderate positive correlation between size and time (r = 0.612). The regression coefficient b = 0.72 is significantly non-zero (*p* = 0.020) and expresses an increase by 0.72 mm in one month.

### 4.5. Identification of PDAC Tissue in the Resected Specimen (Gastric Wall), with the Exclusion of Direct Invasion and Identical Histopathological Pattern with the Primary Tumor

When the gastric wall lesion is diagnosed in a patient with former PDAC resection, the direct invasion of the previous tumor should be excluded. All of our patients had intact gastric serosa macroscopically during the primary resection. Histopathological evaluation of both gastric mucosa and serosa revealed the tumor localization in the muscle layer. The morphologies of the lesion in the gastric wall and the primary pancreatic tumor were identical.

In our cohort, gastric wall metastasis was diagnosed in 19% of patients meeting the inclusion criteria. Artificial seeding is the most appropriate mechanism of origin of such metastases with surprisingly high incidence.

### 4.6. Clinical Relevance of Seeding Metastases of PDAC

Seeding metastases following EUS-FNA are less frequent than after percutaneous FNA, but probably more frequent than has been expected. The clinical significance of seeding metastasis targets a small subgroup of relatively good prognosis cases. The direct impact on seeding into mortality has not yet been proven. Subsequent resection of metastasis is necessary for prolonged disease control with possible influence on the overall survival and morbidity. 

## 5. Conclusions

For patients with the PDAC, who are eligible for upfront surgery, the EUS-FNA is not mandatory and the discussion about abandonment of EUS-FNA in such patients seems to be highly relevant. When neoadjuvant therapy is needed, the EUS-FNA is the method of choice for tissue confirmation. If the needle tract has not been removed during radical surgery (primary tumor location in body/tail of the pancreas), the puncture area is the site of the possible seeding/needle tract metastasis development. In our cohort, there was a remarkably high number of seeding metastases. In the case of solitary seeding metastases, radical resection should always be considered. The seeding PDAC metastases are usually diagnosed during the second year after the primary resection with a usual diameter of 15–30 mm. For patients with the EUS-FNA and subsequent radically resected PDAC, without EUS-FNA needle channel being removed, seeding metastases can be a clinically relevant long-term complication with an estimated incidence of 19% in our cohort.

## Figures and Tables

**Figure 1 biomedicines-10-01392-f001:**
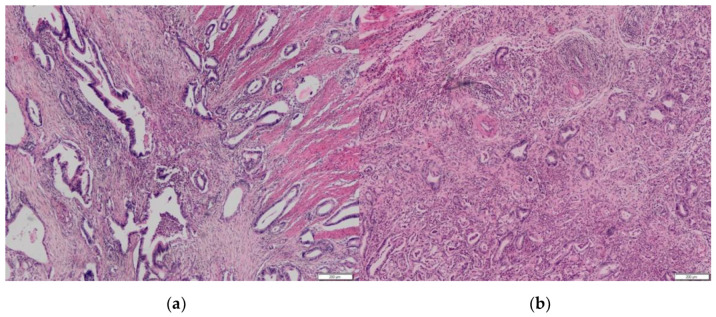
(**a**) Histopathological specimen of pancreatic tissue with well-differentiated ductal adenocarcinoma of pancreatic head. (**b**) Pyloric tissue with well-differentiated ductal adenocarcinoma of pancreatic origin, identic morphology with primary pancreatic tumor. No signs of a primary gastric adenocarcinoma. Hematoxylin/Eosin.

**Figure 2 biomedicines-10-01392-f002:**
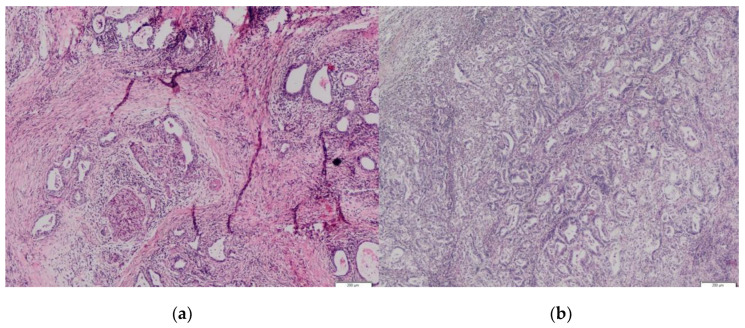
(**a**) Histopathological specimen of pancreatic tail tissue with well- and moderate-differentiated ductal adenocarcinoma. (**b**) Posterior stomach wall tissue with well- and moderate-differentiated ductal adenocarcinoma. Identical morphology with a primary pancreatic tumor. No signs of primary gastric adenocarcinoma. Hematoxylin/Eosin.

**Figure 3 biomedicines-10-01392-f003:**
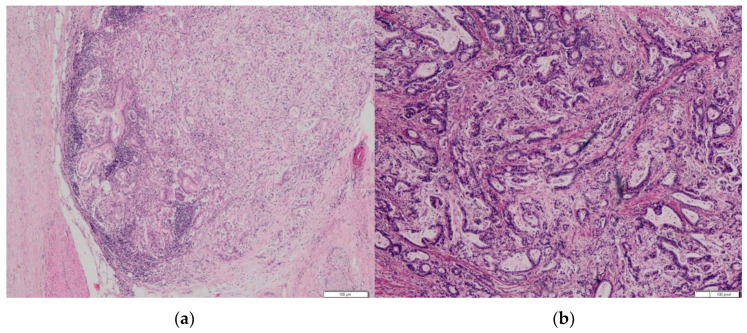
(**a**) Histopathological specimen of pancreatic tail tissue with well- and moderate-differentiated ductal adenocarcinoma. (**b**) Posterior stomach wall tissue with well- and moderate-differentiated ductal adenocarcinoma. Identical morphology with primary pancreatic tumor, no signs of primary stomach adenocarcinoma. Hematoxylin/Eosin.

**Figure 4 biomedicines-10-01392-f004:**
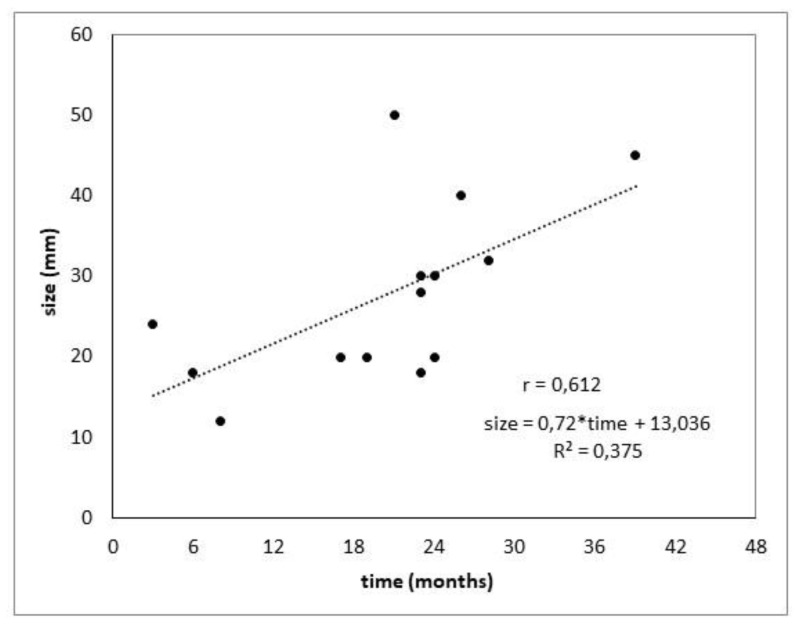
The correlation of time from the EUS-guided FNA and the size of gastric seeding metastasis.

**Table 1 biomedicines-10-01392-t001:** Flow chart of PDAC patients included in the study.

PDAC Patients Radically Resected with Curative Intent	134
Head	107
Excluded (no EUS-FNA)	63
Preoperative EUS-FNA	44
Excluded	43
Resected	1
1-year disease-free interval	1
**Body and Tail**	**27**
Excluded (no EUS-FNA)	12
Preoperative EUS-FNA	15
Excluded	0
Resected	15
1-year disease-free interval	13

**Table 2 biomedicines-10-01392-t002:** Clinical characteristics of patients with proven gastric seeding of PDAC.

	Case 1	Case 2	Case 3
Gender	M	F	F
Age	65	71	75
Presentation	Abdominal pain	Abdominal pain, weight loss	Jaundice
EUS-FNA complication	Haemoperitoneum	0	0
Presentation—Surgery delay	23 M (patient refusal)	1.5 M	1 M
Surgery	Distal pancreatectomy, splenectomy	Distal pancreatectomy, splenectomy	Hemipancreatoduodenectomy s. Traverso
TNM stage, G	pT3N1 M0, G1,R0	pT1 N0 M0, G2,R0	pT3 N0 M0, G3,R0
Oncological therapy	CHT, RT	CHT	CHT
Chemotherapy	5-FU	Gemcitabine 4 cycles	Gemcitabine 6 cycles
Radiotherapy	50.4 Gy	0	0
Gastric lesion presentation	Asymptomatic (PET/CT, EUS)	Asymptomatic (PET/CT, EUS)	Vomiting, weight loss, pylorus obstruction
Postsurgical delay	10 M	26 M	18 M
Serum Ca-19-9 level	1344 kIU/l	796.5 kIU/l	0.6 kIU/l
Diameter	30 mm	25 mm	20 mm
Oncological therapy	Gemcitabine 5 cycles	0	0
Surgery	Distal stomach resection, lymphadenectomy	Distal stomach resection, lymphadenectomy	Pyloric resection, lymphadenectomy
Lymphadenectomy type/positivity	D1 (0 positive)	D1 (3/11 positive)	Peripyloric (2/4 positive)
Single/multiple; serosal involvement	Multiple; no serosal involvement	Single; no serosal involvement	Single; no serosal involvement
Subsequent therapy	DeGramont regimen CHT 7 cycles	Gemcitabine 5 cycles	0
Total survival	56 M	82 M (alive)	28 M
Survival following gastric resection	10 M	54 M (alive)	10 M (no signs of recurrence)

**Table 3 biomedicines-10-01392-t003:** Reported cases of needle tract seeding metastasis after EUS-FNA for pancreatic adenocarcinoma.

Author	Year	Age	Sex	Location	Tumor mm	Passes	Needle G	Treatment	Stage	Recur. M	Size mm	Treatment
Hirooka	2003	57	M	Body	20	3	22	DiPE	T1N0M0	1	Micro	PaGE
Paquin	2005	65	M	Tail	22	5	22	DiPE	T1N0M0	21	50	CHT
Ahmed	2011	79	M	Body	NR	NR	NR	CePE	T2N0M0	39	45	TGE
Chong	2011	55	F	Tail	27	3	22	DiPE	T2N0M0	26	40	NR
Katanuma	2012	68	F	Body	20	4	22	DiPE	T2N0M0	22	NR	NR
Anderson	2013	51	M	Head	50	NR	NR	CHT	NR	NR	10	NR
Ngamruengphong	2013	66	M	Body/Tail	NR	3	22,19	STPE	NR	27	NR	NR
Ngamruengphong	2013	77	F	Tail	40	3	19	DiPE, PaGE	NR	26	NR	NR
Sakurada	2015	87	F	Body	25	NR	22	DiPE	T2N0M0	19	20	PaGE
Minaga	2015	64	F	Body	20	3	22	DiPE	T3N0M0	8	12	PaGE
Tomonari	2015	78	M	Body	20	2	22	DiPE	T3N0M0	28	32	sTGE
Kita	2016	68	F	Body	NR	2	22	RT	NR	4	NR	NR
Yamabe	2016	75	M	NR	30	NR	25	CHT	NR	3	24	CHT
Minaga	2016	72	M	Body	10	NR	NR	DiPE	T1N0M0	24	30	PaGE
Iida	2016	78	F	NR	NR	3	22	DiPE	T3N0M0	6	18	PaGE
Yamanuchi	2018	50	M	Tail	38	2	22	DiPE	T3N0M0	23	28	PaGE
Sakamoto	2018	50	M	Tail	38	2	22	DiPE	T4N1M0	24	20	PaGE
Matsumoto	2018	50	M	Body	35	3	21	DiPE, PaGE	NR	8	NR	PaGE
Matsui	2019	68	F	Body	15	4	19–22	DiPE, PaGE	T1N1M0	1	micro	PaGE
Matsui	2019	70	M	Body	34	1	23	DiPE, PaGE	T3N0M1	4	micro	PaGE
Kawabata	2019	78	F	Body	11	NR	22	DiPE, PaGE	T1N0M0	36	25	PaGE
Sato	2020	83	F	Body	25	2	22	DiPE	T2N1bM0	22	23	PaGE
Rothermel	2020	61	M	Body	37	3	25	DiPE	T3N0M0	42	25	WGE
Okamoto	2020	72	F	Tail	42	5	22	DiPE + PaGE	T3N1M0	-	micro	CHT (Folfirinox)
Yane	2020	66	F	Tail	NR	4	22	DiPE	T3N0M0	18.7	NR	CHT
Yane	2020	78	M	Tail	NR	2	22	DiPE	T3N0M0	26.6	NR	Resection
Yane	2020	86	F	Body	NR	3	22	DiPE	T2N0M0	18.7	NR	Resection
Yane	2020	49	M	Body	NR	4	22	DiPE	T2N0M0	27.8	NR	Resection
Yane	2020	79	F	Body	NR	3	22	DiPE	T1N0M0	36	NR	Resection
Yane	2020	78	F	Body	NR	4	22	DiPE	T1N0M0	34.9	NR	Resection
Lovecek	2022	75	F	Head	25	2	22	PPPDE	T3N0M0	17	20	PaGE
Lovecek	2022	71	F	Body	14	2	22	DiPE	T1N0M0	23	18	PaGE
Lovecek	2022	65	M	Tail	30	4	22	DiPE	T3N1M0	23	30	PaGE

M—male; F—female; G—gauge; NR—not reported; Recur—recurrence; DiPE—distal pancreatectomy; CePE—central pancreatectomy; STPE—subtotal pancreatectomy; PPPDE—pylorus preserving pancreatoduodenectomy; RT—radiotherapy; CHT—chemotherapy; PaGE—partial gastrectomy; WGE—wedge gastrectomy; sTGE—subtotal gastrectomy; TGE—total gastrectomy.

## Data Availability

Not applicable.

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
