# Peer review of "Isolated Gastric Metastases of Pancreatic Ductal Adenocarcinoma following Radical Resection—Impact of Endosonography-Guided Fine Needle Aspiration Tract Seeding"

_biomedicines, 2022, doi:10.3390/biomedicines10061392_

Round 1

Reviewer 1 Report

The topic is relevant, while the risk of tumor-cell seeding after EUS-biopsy is underestimated. What is the recommendation of the authors for the prevention? What kind of measures  do they offer during the index operation? I mean that the site of the biopsy on the gastric wall is scarcely seen during the pancreatic resection.

Author Response

Review – authors reply to reviewer 1:

Reviewer I

The topic is relevant, while the risk of tumor-cell seeding after EUS-biopsy is underestimated. What is the recommendation of the authors for the prevention? What kind of measures do they offer during the index operation? I mean that the site of the biopsy on the gastric wall is scarcely seen during the pancreatic resection.

Reply of the authors:

Thank you very much for the revision, comments and questions.

The prevention of seeding metastasis following EUS-FNA probably could be the resection of the needle tract channel in the stomach during the pancreatic procedure. The technical problem is the marking of the needle tract. We suppose a sufficient and adequate approach should be a thorough follow up with a targeted gastric wall check.

In the case of upfront surgery, it seems to be more efficient to avoid EUS-FNA.

In case of neoadjuvant therapy and necessity to perform EUS FNA, strict CT and endoscopic follow up with gastric wall check using gastroscopy +/- EUS should be used.

Reviewer 2 Report

Introduction

Major comments:

  • For the purpose of analysis of a homogeneous population, are the authors able to say, whether adenocarcinomas of the head of the pancreas were trans-gastric biopsied  ?

In case of trans-duodenal biopsy, in my opinion, it is not justified to include cephalic localizations.

Materials and methods:

Major comments:

  • Flowchart of the included population with inclusion criteria and reason for exclusion
  • It is quite surprising to see the survival calculated from the date of diagnosis.

Wouldn't it be more relevant to do it from the date of diagnosis?

  • Would it be possible to know the status R at the time of surgery of the primary pancreatic lesion?

Minor comments :

The JASPAC and PRODIGE24 studies have changed the survival and guidelines for treatment of pancreatic adenocarcinoma. There are few data on this topic.

  • Even though it is a serious pathology, all patients died in relation to their metastatic pathology?

Can the authors specify the morbidity and mortality of GSM surgery?

Is there a real place for lymphadenopathy? How many N+? In this case, wouldn't it be a disseminated metastatic pathology, which would contraindicate surgery?

Discussion and results:

  • Since 2020 other article have been published.

To my knowledge the authors have not taken into consideration:

- Tang J; J Int Med Res 2021

-Okamoto T; Clin J Gastro 2020

- Kojima H; WJG 2021

- Jamaguchi H; Inter Med 2020

Author Response

Reviewer II

Reply of the authors:

Thank you very much for your extended review and recommendation. This is our point-to-point reply to Your comments.

Introduction

Major comments:

For the purpose of analysis of a homogeneous population, are the authors able to say, whether adenocarcinomas of the head of the pancreas were trans-gastric biopsied? In the case of trans-duodenal biopsy, in my opinion, it is not justified to include cephalic localizations.

For the purpose analysis: Of course yes, the pancreatic head tumors are biopsied routinely via duodenum and the needle tract is regularly removed during the resection procedure. In the case of standard manner EUS-FNA performance, the seeding should not occur. In the discussion, we mention the unique and to date unpublished case of GSM following DPE. We concluded, that EUS FNA, in this case, was not done in a standard manner and the needle tract was directed via pylorus. In the medical record, we are not able to identify the detailed proper site of the needle tract. In the discussion, we mention the case of suspected hematogenic metastasis in the pyloric region as a possible secondary cause.

Materials and methods:

Major comments:

  • Flowchart of the included population with inclusion criteria and reason for exclusion

A flowchart has been added.

  • It is quite surprising to see the survival calculated from the date of diagnosis. Wouldn't it be more relevant to do it from the date of diagnosis?

The date of diagnosis (primary) in the first case was only on a CT scan. The delay was due to the patient's decision (even in clinical suspicion). This situation should lead to the point of setting the diagnosis as primary datation of survival.

  • Would it be possible to know the status R at the time of surgery of the primary pancreatic lesion? 

R status was added to the list of GSM patients.

Minor comments :

The JASPAC and PRODIGE24 studies have changed the survival and guidelines for the treatment of pancreatic adenocarcinoma. There are few data on this topic.

  • Even though it is a serious pathology, all patients died in relation to their metastatic pathology?

The first presented patient died in relation to his metastatic pathology, the second one is still alive without sign of recurrence, third died due to comorbidities without clinical sign of recurrence (Tab. 2)

Can the authors specify the morbidity and mortality of GSM surgery?

Morbidity and mortality following GSM surgery were 0.

Is there a real place for lymphadenopathy? How many N+? In this case, wouldn't it be a disseminated metastatic pathology, which would contraindicate surgery?

Patient 1. Only regional/perigastric lymphadenectomy, no systematic od DII lymphadenectomy, no positive lymph node.

Patient 2. 11 lymph nodes, 3 positive. Surgery performed, still alive, PETCT XII/2021 – no sign of recurrence.

Patient 3. Pyloric resection, regional lymphadenectomy – 4 nodes, 2 positive.

The experience supports the indication of resection even in the case of regional lymphadenopathy. The limiting situation for radical therapy is serosal involvement.

Discussion and results:

  • Since 2020, other articles have been published. To my knowledge the authors have not taken into consideration:

- Tang J; J Int Med Res 2021

Yang J, Yan Y, Zhang S, Lv Y. Gastric metastasis from pancreatic cancer characterized by mucosal erosion: a case report and literature review. Journal of International Medical Research 2021:49(4)1-6.

… we suppose Yang Jie et al. Gastric metastasis from pancreatic cancer characterized by mucosal erosion: a case report and literature review. Journal of International Medical research 2021:49(4)1-6.

The citation is added to the text in the discussion in the section 4.2

-Okamoto T; Clin J Gastro 2020

Okamoto T, Nakamura K, Takasu A, Kaido T, Fukuda K. Needle tract seeding and abscess associated with pancreatic fistula after endoscopic ultrasound-guided fine-needle aspiration. Clin J of Gastroenterology 2020;13:1322-1330

We have added this case to the list of PDAC GSM treated with curative intent so far.

Yane K, Kuwatani M, Yoshida M et al.Non-negligible rate of needle tract seeding after endoscopic ultrasound-guided fine-needle aspiration for patients undergoing distal pancreatectomy for pancreatic cancer. Digestive Endoscopy 2020;32:801-811.

This publication was added to the previously published group.

- Kojima H; WJG 2021

This publication is non-relevant due to presented peritoneal dissemination (the paper is targeted at gastric metastasis)

- Jamaguchi H; Inter Med 2020

This publication is not relevant – different pathology – SPN.

Round 2

Reviewer 2 Report

None